engineering geology/energy/geology

overlying remnant pillars, multiseam mining, ground instability, risk assessment, longwall mining

**Authors for correspondence:**
Yong Li
e-mail: yong.li@cqu.edu.cn
Ruikai Pan
e-mail: panrk@cqu.edu.cn

# Assessment of ground instability risk of lower seam longwall panels during crossing overlying remnant pillars

Ruikai Pan[1,2], Yong Li[1,2], Hui Wang[1,2], Youlin Xu[3] and Hongyun Yang[4]

[1]State Key Laboratory of Coal Mine Disaster Dynamics and Control, and [2]School of Resource and Safety Science, Chongqing University, Chongqing 400044, People's Republic of China
[3]College of Mining Engineering, Guizhou Institute of Technology, Guiyang 550003, People's Republic of China
[4]State Key Laboratory of Mountain Bridge and Tunnel Engineering, College of Civil Engineering, Chongqing Jiaotong University, Chongqing 400074, People's Republic of China

RP, 0000-0003-0230-2400; YL, 0000-0003-2860-2313

One of the most challenging safety problems is ground instability during crossing overlying remnant pillars (CORP). Ground instability not only causes injury to miners or fatalities, but also leads to interruptions in the mining operations and breakdowns in equipment. In this paper, 12 major parameters influencing the ground instability were firstly determined based on extensive international experience associated with CORP. The consequences of the ground instability were then assessed in terms of miners' health and financial losses. Afterwards, a practical method to assess the ground instability risk of lower-seam longwall panels during CORP was developed based on its probability and consequence. Finally, this method was successfully used to determine the best scheme for CORP of LW10-103 at Mugua coal mine. The main advantage of this method is that it enables mining engineers to easily use international experience for assessing the risk of ground instability and selecting reasonable supports during CORP.

## 1. Introduction

As the intensity of coal mining increases, coal resources at shallower depths continue to decrease, so there is a need to extract lower seams in many mining areas [1,2]. The lower-seam longwall workings can inevitably encounter the remnant pillars

left in the upper seams. In some cases, the longwall workings will cross the overlying pillars in order to maintain the continuity of production and increase the recovery of coal resources [3]. However, crossing overlying remnant pillars (CORP) increases the risk of ground control. The risk of ground control can include roof falls, rib spalling and floor heave, which can result in injury to miners, fatalities, interruptions in the mining operations, breakdowns in equipment, and so forth [4–6]. Therefore, a proper method for the assessment of ground control risk is of critical importance in order to enable engineers to correctly estimate the risk of CORP and properly plan out mining [7].

The most challenging problem encountered during CORP is the high stress concentration generated by the remnant pillars in the overlying seams. Field measurements and numerical modelling studies have shown that the vertical stress concentrations caused by the remnant coal pillars can affect the interburden within a distance of 120–200 m below it [8–10]. The vertical stress concentration under remnant pillars is a function of the abutment angle, pillar width, pillar type and interburden depth [11,12]. In addition to high stress concentration from remnant pillars, other technical factors affecting the ground control of the face and roadway during CORP have been identified. Research by Mark [13] and Zipf [14] revealed that the most difficult interactions have been associated with remnant pillars oriented parallel with the longwall face. Xu and co-workers [15] and Ju et al. [16] reported that it is often necessary to use powered supports with high working resistance and increase the scope and intensity of reinforced supports in the roadways during CORP in China. Some factors that were not found to be statistically significant included the time lag between mining the two seams and the interburden competence [17].

Although there are few studies on the risk assessment of CORP, the above engineering experience provides a good understanding and basis of the risk assessment of CORP. In fact, many scholars have researched the risk assessment of ground control for several other types of mining activities. For example, a practical method for assessing the risk of roof falls in longwall panels is presented based on the empirical methods and expert techniques in Poland [18]. Similar studies are conducted to judge the overall risk of pillar recovery operation based on available roof fall data or expert experience [19–22]. These mature risk assessment techniques can be considered for the risk assessment of CORP.

In the current work, the technical background to CORP is firstly described in detail. Secondly, based on the review of the extensive related literature and expert experience, the main factors in the ground control during CORP are introduced, and the role of each factor is described. Thirdly, a semi-quantitative method for evaluating the risk of ground control is proposed by combining the probability of risk occurrence and the consequence of risk. Finally, the successful application of this method to Mugua mine is introduced.

It should be noted that the types of remnant pillars vary with the mining method, leading to different effects on the lower-seam mining. The scope of this study is limited to the case where both working faces of the upper and lower coal seams are longwalls.

# 2. Background

## 2.1. Three basic types of crossing overlying remnant pillars

In practice, a range of factors such as dip, water management and geological features often prevent multiseam workings from being orientated in the same direction [23]. As a result, the lower-seam longwalls need to cross these remnant pillars for continuing mining. Another situation is that the upper and lower working faces have the same advancing direction. At this time, entries will be staggered from the overlying remnant pillars to ensure the stability of the roadway. However, this arrangement also means that the effect of the stress concentration of the overlying remnant pillars will be manifested in the bottom seam longwall face [4,5]. In summary, the lower-seam longwall will inevitably be affected by the overlying remnant pillars during the extraction.

For longwall extraction, the remnant pillars can generally be classified as either: [24]

(i) isolated pillar that is subjected to two overlapping abutments; or
(ii) gob–solid boundary that carries a single, distributed abutment load.

Figure 1 displays a characteristic sketch of the three basic types of CORP. Figure 1a shows the longwall in the lower seam crossing the isolated pillars. The whole process involves entering the pillar line, mining under the pillar and exiting the pillar. Figure 1b,c depicts the lower-seam longwall crossing the gob–solid boundary from the gob to the solid and from the solid to the gob, respectively.

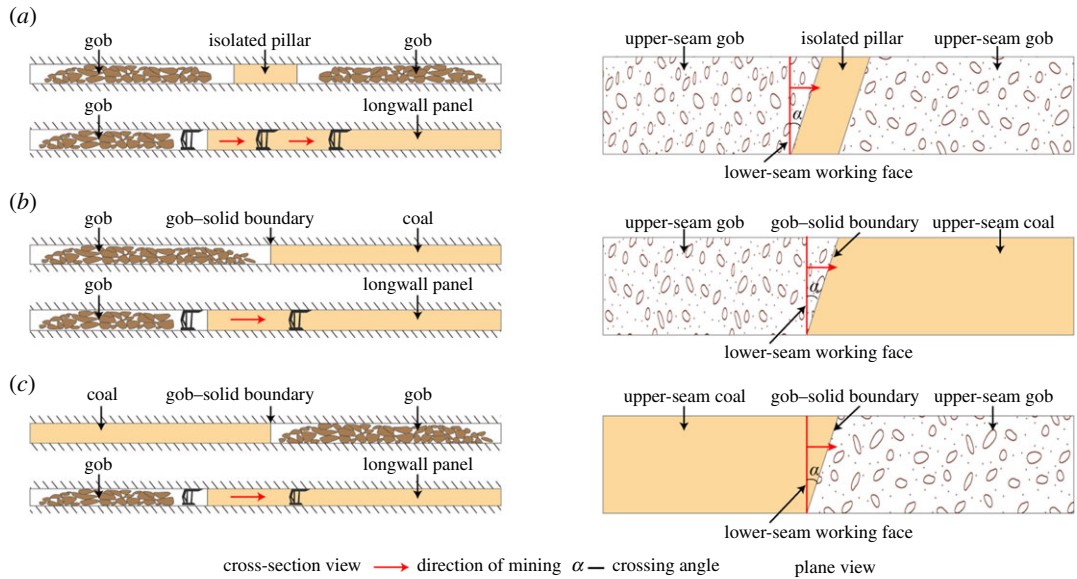

**Figure 1.** A characteristic sketch of three basic types of CORP: (*a*) crossing the isolated pillar; (*b*) entering the solid coal; (*c*) entering the gob.

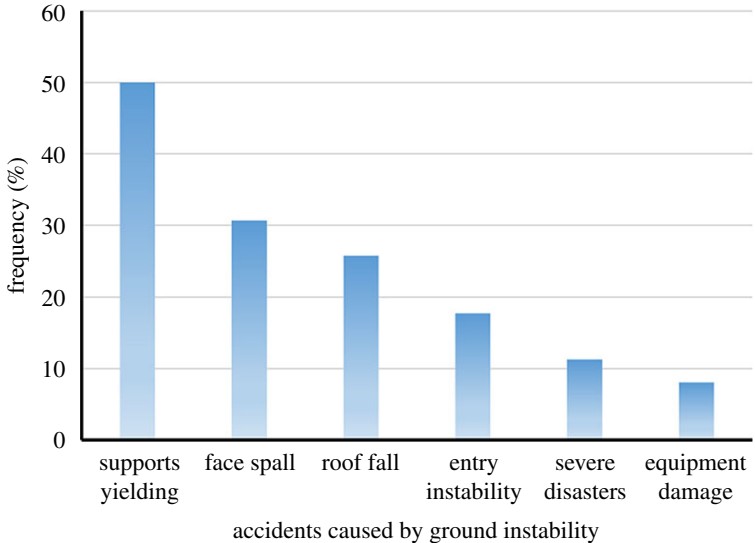

**Figure 2.** Various types of accidents caused by ground instability.

The crossing angle is referred to as the angle between the lower-seam longwall face and the longitudinal direction of the upper-seam pillar, corresponding to $\alpha$ in figure 1. When $\alpha$ is approximately equal to zero, the longwall face is orientated perpendicular to the pillar, while when $\alpha$ is approximately equal to 90°, the lower-seam longwall panel advances along the longitudinal direction of the upper-seam pillar.

## 2.2. Ground instability associated with crossing overlying remnant pillars

Figure 2 illustrates various types of accidents caused by ground instability in China coal mines over the years [6,15,16]. It can be observed that the yielding of powered supports is the first common type of accident, and face spalling is ranked second. Roof fall and entry instability are the other types of accidents having a relatively high share. One of the most striking features of the accidents associated with CORP is the dynamic failure. For example, several types of dynamic events, including the ejection of coal from the face, the rapid rate of yielding and closure of powered supports and violent roof collapse, often occur. These ground instability risks can be avoided to a certain extent by conducting risk assessment in advance.

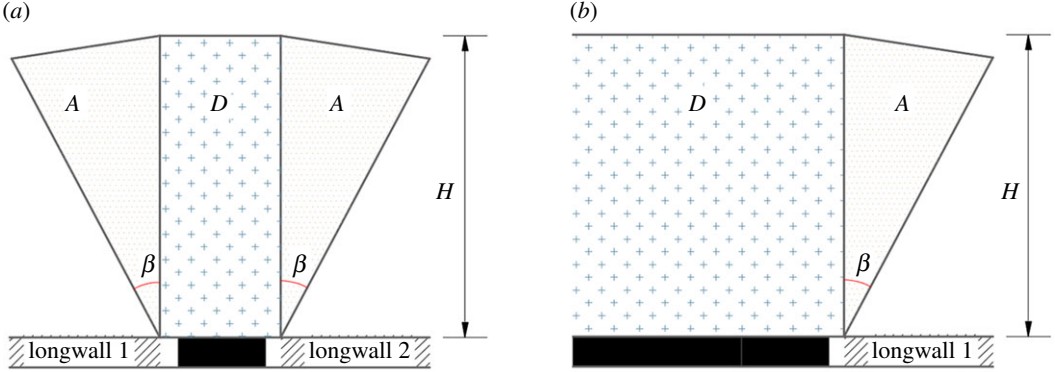

*A*, abutment load;    *β*, abutment angle;    *D*, dead weight;    *H*, depth of cover;

**Figure 3.** Components of the load in two types of coal pillars: (*a*) isolated pillar; (*b*) gob–solid boundary.

# 3. Parameters contributing to ground instability

The safety of CORP depends on the variety of geological and technical parameters which can be divided into three groups according to the mechanism of ground instability:

(i) loading parameters that describe the magnitude and distribution of the load applied to the remnant pillar structure;

(ii) transfer parameters that define the transfer capacity of loading through the interburden to the lower-seam longwall panel; and

(iii) designing parameters that involve the geometric dimensions of the lower-seam longwall panel and the way it crosses the overlying coal pillars.

The parameters (marked IC—instability cause) used in this work are based on prior engineering investigations and experiments in the major coal producing countries. Each parameter divides individually into several subgroups according to the risk of CORP. A risk value is defined to each of these subgroups as an index which represents the risk level of CORP. The risk value can be a number between 0 and 4; value 0 represents no risk condition, 1 is low risky condition, 2 is medium risky condition, 3 is high risky condition and 4 is extreme risky condition.

## 3.1. Loading parameters

At the completion of longwall mining, the load can be separated into two parts: the dead weight of overburden and the abutment load that is not carried by the goaf but is transferred to the pillars, as shown in figure 3. The dead weight and the abutment load can be evaluated by depth and abutment angle, respectively.

### 3.1.1. Depth of cover (IC1)

The increased depth of cover results in high virgin stress levels in the rock mass, i.e. a higher dead weight both vertically and horizontally [25]. Therefore, it is difficult for longwalls to achieve sufficient stability at a higher depth. An exception arises when mining at shallow depths, where the surface effect can lead to overburden failure in a brittle and unpredictable mode [16]. In China, this situation occurs when the depth of cover is less than 150 m [15,16]. Thus, the depth of cover is divided into four categories: extreme, when it is less than 150 m or more than 600 m; high, when it is between 450 and 600 m; medium, when it is between 300 and 450 m; and low, when it is between 150 and 300 m.

### 3.1.2. Abutment angle (IC2)

The abutment load carried by the pillar can be characterized by the abutment angle, *β*, as depicted in figure 3. It can be seen that the abutment load increases with the increasing abutment angle [26]. The abutment angle should not be considered as a physical reality, but rather as a conceptual model that represents the amount of mining-induced loads on the abutments [26]. It has been suggested that the

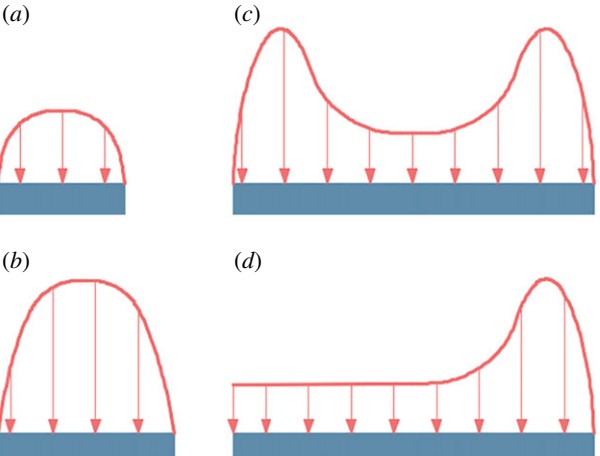

**Figure 4.** Stress distribution on different coal pillars: (*a*) yielded pillar; (*b*) critical pillar; (*c*) wide pillar; (*d*) gob–solid boundary.

abutment angle correlates with the geological strength and bridging capacity of the overlying strata. However, no evidence has been collected to prove this assertion [27]. In the USA, the average number of 21° is accepted as the standard abutment angle [26]. On the basis of reviewing the literature, Peng [28] concluded that the abutment angle ranges from 5° to 35°. Field measurements showed that the value changes from 5° to 40° [29]. Therefore, the abutment angle is divided into three categories: high, when it is between 25° and 40°; medium, when it is between 15° and 25° and low, when it is between 0° and 15°.

### 3.1.3. Pillar type (IC3)

The load acting on the coal pillar is also related to the type of coal pillar. According to the load distribution characteristics, four different types of coal pillars may be defined [30,31]:

  (i)  yielded pillars that yield but remain stable and carry relatively small loads; they are generally 4–10 m wide (figure 4*a*);
 (ii)  critical pillars that are highly stressed throughout and are generally 10–50 m wide (figure 4*b*);
(iii)  wide pillars that carry a larger total load, but because that load is distributed over a much larger area, its effect on the lower seam is less notable; these are generally 50–100 m wide (figure 4*c*); and
 (iv)  gob–solid boundaries that have a localized high stress at the edge of solid coal (figure 4*d*).

Generally, small yielded pillars pose less significant hazards than the other types of pillars. As gob–solid boundaries only carry one stress abutment, they ideally have a less adverse effect compared to the two others. On the other hand, critical pillars can be considered more dangerous than wide coal pillars owing to their higher stress concentration. Therefore, the types of pillars can be divided into four subgroups: extreme, when it is a critical pillar; high, when it is a wide pillar; medium, when it is a gob–solid boundary and low, when it is a yielded pillar.

## 3.2. Transfer parameters

### 3.2.1. Interburden thickness (IC4)

The interburden thickness is one of the most important parameters because the stress concentration beneath remnant pillars decreases as the interval between the coal seams increases [11,24]. Another reason why interburden is important is that the thicker the interburden is, the less likely the concentrated load acts on the working face when the working face passes through the coal pillar [32]. As shown in figure 5, the longwall face is located on the boundary of the coal pillar, and the stratum movement can affect the overlying pillar when the interburden is thin, which may cause an extensive roof fall along the break line, leading to the yielding of the powered supports. In the case of a thicker interburden, the movement of the stratum completely disturbs the upper-seam coal pillar when the working face is far away from the coal pillar. At this time, the dynamic load caused by the pillar

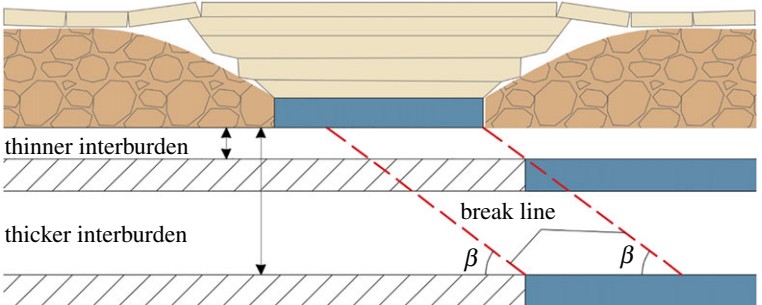

**Figure 5.** Strata movement under different interburden conditions: (*a*) thinner interburden; (*b*) thicker interburden.

instability acts on the gob material, which has a little impact on the longwall face, thereby reducing the probability of roof fall.

If the mining height of the lower seam is $M$, and the interburden thickness is $t$, the interburden can be divided into four subgroups [21]: extreme, when $t$ is less than $4M$; high, when $t$ is between $4M$ and $10M$; medium, when $t$ is between $10M$ and $24M$; and low, when $t$ is between $24M$ and $60M$.

### 3.2.2. Interburden geology (IC5)

When mining under coal pillars, the broken interburden may bridge, thereby dampening the downward load transfer [23]. Under the premise of being able to fracture, the harder and thicker the rock layer, the higher the degree of the bridge and thus the better the dampening effect on the coal pillar load. In general, the percentage of hard rock in the interburden (sandstone, limestone, etc.) is used to quantify this property of interburden, that is, the higher the percentage is, the stronger the ability to bridge is [3]. The percentage of competent rock in the interburden was calculated based on boreholes by summing the total hard rock and then dividing it by the total interburden thickness [17]. Generally, the rock with a uniaxial compressive strength higher than 30 MPa is considered to be hard rock [33]. Hard rock percentage ($p$) is divided into three categories based on international knowledge: high, when $p$ is less than 40%; medium, when $p$ is between 40 and 70% and low, when $p$ is between 70 and 100%.

### 3.2.3. Overlying massive strata (IC6)

Safe mining under pillars does not mean preventing roof caving but ensures that it should occur uniformly and constantly at an appropriate span. One of the most important factors affecting roof caving is whether there is a massive stratum such as sill over the panel [20]. Massive strata often tend to cantilever a considerable distance out into the gob, resulting in excessive abutment stress in the working face. After achieving a critical span, they break suddenly and dynamically, causing a roof fall and damaging hydraulic supports. Field observations indicate that when the rock layer is thicker and vertically closer to the working face, the adverse effect of the massive strata on the working face is greater [34]. Based on Anderson's works [35], a massive stratum is divided into three categories: extreme, when the massive stratum is within 20 m above the face; high, when it is within 20–60 m above the face and medium, when it is more than 60 m above the face.

## 3.3. Designing parameters

### 3.3.1. Mining height (IC7)

Mining height affects the ground control of the working face through three aspects: the height of the caving zone, the stiffness of the coal face and the support, and the exposure of the discontinuities on the wall [23]. Firstly, the greater the mining height, the greater the height of the caving zone, and therefore, the thicker the strata bed resting on the longwall face supports. Secondly, as mining height increases, the stiffness of both the coal face and the powered supports is reduced, resulting in increased roof convergence and rib deformation. Thirdly, there is a higher likelihood that discontinuities are exposed in the coal wall at higher mining heights, thereby increasing the risk of face spall [36]. Therefore, mining height is divided into four categories according to the latest coal

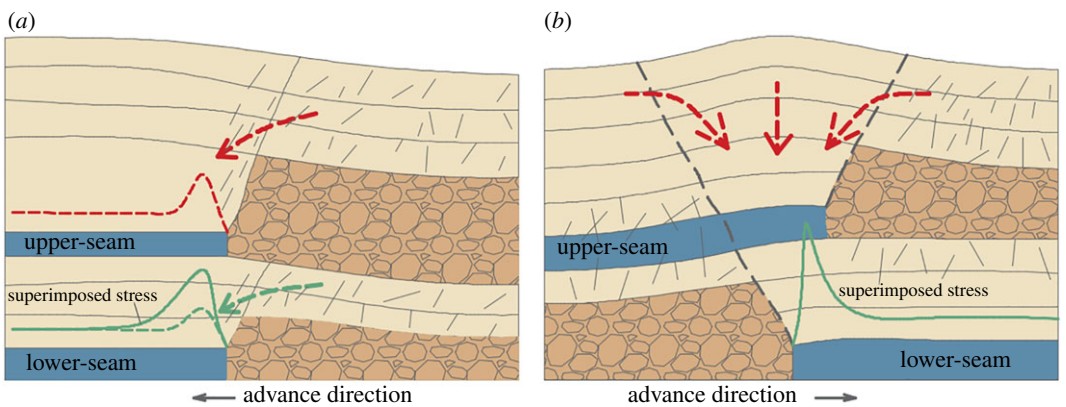

**Figure 6.** Interaction between upper-seam gob–solid boundary and lower-seam longwall panels: (*a*) advancing from the gob to the solid; (*b*) advancing from the solid to the gob.

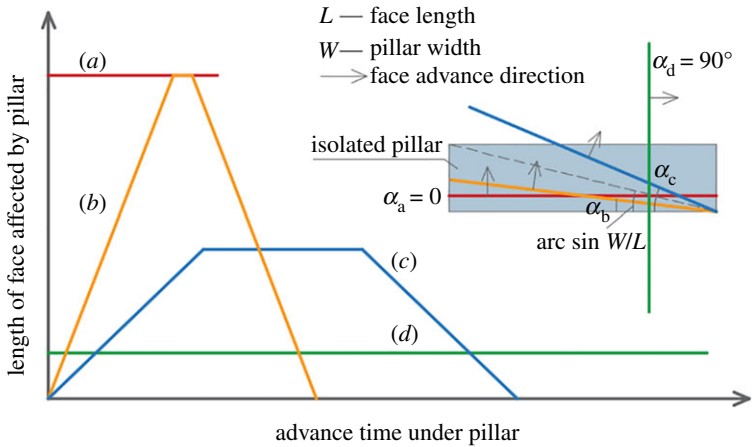

**Figure 7.** Changes in length of the working face affected by coal pillars at different crossing angles.

mining technology: extreme, when it is more than 5.5 m; high, when it is between 3.5 and 5.5 m; medium, when it is between 1.3 and 3.5 m; low, when it is less than 1.3 m.

### 3.3.2. Direction of mining (IC8)

As mentioned in §2.1, the mining direction is mainly for the gob–side boundary, including two possibilities: from the gob to the solid and from the solid to the gob. For the case of crossing an isolated coal pillar, there is only one mining configuration, i.e. from the gob to the solid and then to the gob. For crossing gob–solid boundaries, a large number of studies have indicated that it is best to retreat from the gob towards the solid [17,23,32]. Figure 6*a* shows the working face advancing from the gob to the solid. When the working face approaches the coal pillar, its abutment stress is approximately twice as large as the peak value of single coal seam mining because of the superimposition of two stress fields created by workings in two adjacent seams. Figure 6*b* depicts the working face advancing from the solid to the gob. Owing to the structural damage to the solid coal, the proportion of the intact coal gradually decreases, giving rise to a great increase of the abutment stress in the lower mining face. Additionally, the findings of several studies confirm that isolated remnant pillars cause more intense interactions and problems than the gob–solid boundaries [13,17,32]. Therefore, the mining direction is divided into three categories: extreme, when the face crosses the isolated pillar; high, when the face advances from the solid to the gob; and low, when the face advances from the gob to the solid.

### 3.3.3. Crossing angle (IC9)

Crossing angle is also an important parameter because it controls how the pillar load acts on the working face during CORP. Figure 7 illustrates the changes in the length of the working face directly loaded by

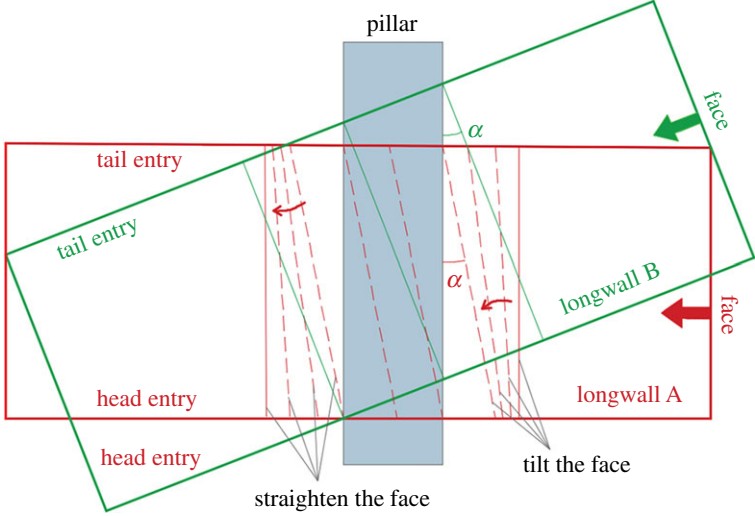

**Figure 8.** Two forms of the working face obliquely crossing the coal pillar.

coal pillars at different crossing angles as the working face advances. Large field investigations confirm that the most difficult interactions have been associated with the layouts where the working face advances perpendicular to the coal pillar, that is, the crossing angle equals zero. In this situation, the longwall face is subjected to a high abutment stress along its full length as it passes beneath the pillars as depicted by line (*a*) [3,15]. The best situation arises when the working face of the upper and lower seams are parallel to each other, i.e. a crossing angle of 90°. Such an arrangement minimizes the length of the face loaded by the pillar as indicated by line (*d*) [30]. The mode represented by line (*b*) is inferior to the one indicated by line (*a*) as the full-length working face is loaded by the pillar during some periods in line (*b*). If the crossing angle is $\alpha$, the face length is $L$, and the pillar width is $W$, the angle can be divided into four subgroups: extreme, when $\alpha$ is zero; high, when $\alpha$ ranges from zero to arcsin ($W/L$); medium, when $\alpha$ ranges from arcsin ($W/L$) to 90°; and low, when $\alpha$ equals 90°.

For the gob–solid boundaries, $W$ can be considered as the lateral extent of the side abutment zone. Peng & Chiang [37] found that the maximum extent of the abutment load is a function of the depth, as given by the below equation [37]:

$$W_{\mathrm{SAZ}} = 2.84\sqrt{3.3H}, \tag{3.1}$$

where $W_{\mathrm{SAZ}}$ is the lateral extent of the side abutment zone and $H$ is the depth of the cover.

Owing to the lack of scientific understanding of CORP in the early days, engineers arranged the upper-seam and lower-seam longwall faces perpendicular to each other. As a result, the risk can only be reduced by tilting the working face at a small angle (longwall *A*), as shown in figure 8. The risk reduction by this way is much smaller than that by tilting both the entries and working face (longwall *B*).

### 3.3.4. Powered supports in the face (IC10)

The longwall face is supported by powered supports, the main function of which is to provide a safe working environment as the coal is extracted. The support performance of hydraulic supports is usually described by the support intensity [23]. Roof support intensity, also referred to as support density, is defined as the thrust offered by the support to the area of the roof supported [38]. This area is measured from the coal face to the edge of the last supporting element on the gob side of the face. The total thrust is based on the sum of the nominal yield loads of all the hydraulic support elements in the system. In order to compare the performance of the supports, the safety factor of a powered support is defined as

$$s = \frac{P_{\mathrm{S}}}{P_{\mathrm{L}}}, \tag{3.2}$$

where $P_{\mathrm{S}}$ is the support intensity and $P_{\mathrm{L}}$ is the strata load on the support. Peng [28] recommended that $P_{\mathrm{L}}$ should commonly take the weight of the strata, the height of which is eight times the mining height. Therefore, the safety factor of the support is divided into three categories: extreme, when *s* is smaller

than 1.0; high, when $s$ ranges from 1.0 to 1.3; medium, when $s$ ranges from 1.3 to 1.6; and low, when $s$ is higher than 1.6.

### 3.3.5. Hydraulic props in entry (IC11)

During the longwall panel retreat period, hydraulic props are commonly used to support the roof ahead of the working face. By increasing the density and expanding the distance of hydraulic props, the stability and safety of the entrance during CORP can be ensured. Under normal geological conditions, hydraulic props are installed at least within 20 m ahead of the working face in China [39]. Therefore, the support increasing ratio of hydraulic props can be defined as

$$\lambda = \frac{L_P}{L_N} \cdot \frac{R_P}{R_N} \cdot \frac{S_N}{S_P}, \tag{3.3}$$

where $L_P$ and $L_N$, $R_P$ and $R_N$, and $S_P$ and $S_N$ are the support distance, the number of rows and the row spacing of the props under remnant pillar and normal conditions, respectively.

Based on this equation, a larger $\lambda$ corresponds to the higher support intensity of props. When $\lambda$ is equal to 1, it means that the support intensity has not improved. According to the support increasing ratio, the support of hydraulic props can be divided into four subgroups: extreme, when $\lambda$ equals 1; high, when $\lambda$ is between 1 and 2; medium, when $\lambda$ is between 2 and 4; and low, when $\lambda$ is higher than 4.

### 3.3.6. Grouting of the surrounding rock (IC12)

The surrounding rock here mainly includes the coal wall and the roof of the working face. During CORP, the surrounding rock is subjected to a high abutment pressure, causing the development of cracks on the surface and inside of the rock. If these cracks are not well controlled, the surrounding rock will break or even become unstable, leading to a roof fall and face spall [23]. An important way to solve this issue is the grouting of the fractured rock mass. Grouting substances including chemical and cementitious materials can fill the mining-induced cracks and voids and bind the fractured rock properly, thereby turning the loose weak rock mass into a strong solid material [40]. Therefore, the grouting of the surrounding rock can be divided into two subgroups: high, when the surrounding rock is broken but not grouted; and low, when the surrounding rock is broken and grouted.

# 4. Assessment of the ground instability risk

Risk is defined as the chance of the occurrence of unwanted events that will have adverse effects on purpose [41]. The probability of an event and its consequence are combined to determine the level of risk. Ground instability during CORP continues to be one of the greatest geotechnical risks faced by underground coal miners and causes many losses, injuries or fatalities [13]. Therefore, the ground instability risk can be defined as

$$R_{GI} = P \times C, \tag{4.1}$$

where $R_{GI}$ is the risk of ground instability, $P$ represents the probability of ground instability occurrence and $C$ stands for the consequence of ground instability.

In order to assess the risk of events such as roof fall, quantitative, semi-quantitative or qualitative methods can be used [19]. For quantitative risk assessment, the probability is determined based on available historical data and statistical analysis, and the consequence is determined using measurable factors such as monetary unit, fatality or lost time. When the probability and consequence cannot be specified exactly and are determined subjectively, this method is called semi-quantitative or qualitative risk assessment. In these methods, the probability and consequence are determined based on judgement, experience and experts' opinions.

Owing to lack of sufficient effective cases for quantitative risk assessment, this paper employs a semi-quantitative method to assess the risk of ground instability during CORP. In fact, this method has been used by some scholars to evaluate the roof fall risk in longwall mining and retreat mining [18,21]. In the following section, the methods for quantifying probability and consequences are described in detail.

**Table 1.** Loading parameters influencing ground control risk during CORP.

| parameters | probability factor (PF) | weight |
|---|---|---|
| IC1: *depth of cover* (m) | | |
| less than 150 | 4 | 0.1033 |
| between 150 and 300 | 1 | |
| between 300 and 450 | 2 | |
| between 450 and 600 | 3 | |
| more than 600 | 4 | |
| IC2: *abutment angle* (°) | | |
| less than 15 | 1 | 0.02 |
| between 15 and 25 | 2 | |
| between 25 and 40 | 3 | |
| more than 40 | 4 | |
| IC3: *pillar type* | | |
| yielded pillar | 1 | 0.0953 |
| critical pillar | 4 | |
| wide pillar | 3 | |
| gob–solid boundary | 2 | |

**Table 2.** Transfer parameters influencing ground control risk during CORP. (*D*, distance from the coal seam, m.)

| parameters | probability factor (PF) | weight |
|---|---|---|
| IC4: *interburden thickness* (m) | | |
| less than 4*M* | 4 | 0.1694 |
| between 4*M* and 10*M* | 3 | |
| between 10*M* and 24*M* | 2 | |
| between 24*M* and 60*M* | 1 | |
| more than 60*M* | 0 | |
| IC5: *interburden geology/p* | | |
| less than 40 | 3 | 0.0788 |
| between 40 and 70 | 2 | |
| between 70 and 100 | 1 | |
| IC6: *overlying massive strata/D* | | |
| not present | 0 | 0.0642 |
| present/less than 20 | 4 | |
| present/ between 20 and 60 | 3 | |
| present/more than 60 | 2 | |

## 4.1. Probability of ground instability

As mentioned in §3, 12 parameters affecting the ground instability during CORP were determined. Each parameter is divided into several subgroups assigned a proper probability factor (tables 1–3). It should be noted that these parameters and the probability factors of each subcategory were determined by consulting 10 experts. The invited experts evaluated and improved the initial index system. For example, grouting was not included in the initial index system, but was added based on the suggestions of some experts.

**Table 3.** Design parameters influencing ground control risk during CORP.

| parameters | probability factor (PF) | weight |
|---|---|---|
| IC7: *mining height* (m) | | |
| less than 1.3 | 1 | 0.1196 |
| between 1.3 and 3.5 | 2 | |
| between 3.5 and 5.5 | 3 | |
| more than 5.5 | 4 | |
| IC8: *direction of mining* | | |
| from gob to solid | 1 | 0.08 |
| from solid to gob | 3 | |
| both involved | 4 | |
| IC9: *crossing angle* (°) | | |
| 0 | 1 | 0.1303 |
| between 0° and arcsin($W/L$) | 2 | |
| between arcsin($W/L$) and 90 | 3 | |
| 90 | 4 | |
| IC10: *powered support/s* | | |
| less than 1 | 4 | 0.0538 |
| between 1 and 1.3 | 3 | |
| between 1.3 and 1.6 | 2 | |
| more than 1.6 | 1 | |
| IC11: *hydraulic props/$\lambda$* | | |
| 1 | 4 | 0.0462 |
| between 1 and 2 | 3 | |
| between 2 and 4 | 2 | |
| more than 4 | 1 | |
| IC12: *grouting of surrounding rock* | | |
| not broken | 0 | 0.0391 |
| broken and not grouted | 3 | |
| broken and grouted | 1 | |

Since each parameter has a different effect on the ground instability, it is necessary to determine the weight of each parameter. To this end, the analytic hierarchy process was employed. A total of 30 experts were consulted in this study. Experts compare and score the relative importance of every two parameters. Then, the final weight is obtained by performing statistical calculations on the experts' results (see tables 1–3). This approach is explained in detail in Yang's research [42]. The detailed survey content is shown in appendix A. There are also two options in the questionnaire, namely the expert's judgement basis for the problem and the degree of familiarity with the problem.

Among the 30 experts invited in this survey, 20 experts have senior professional titles, and the rest have intermediate professional titles. All the questionnaires sent to the experts were effectively recovered, indicating that the experts paid great attention to the content of the consultation. In addition, experts are familiar with the field of multiple seam mining, and their judgements are mostly based on theoretical analysis and practical experience, so the consultation results were credible.

By introducing the probability factor and the weight of each parameter into equation (4.2), the probability of ground instability during CORP can be calculated as

$$P = \left[ \left( \sum_{i=1}^{12} \mathrm{PF}_i \times W_i \right) \middle/ \left( \sum_{i=1}^{12} \mathrm{MPF}_i \times W_i \right) \right] \times 100, \tag{4.2}$$

**Table 4.** Ground instability probability assessment.

| assessment of ground instability cause | ground instability probability |
|---|---|
| 1–35 | low |
| 35–55 | medium |
| 55–75 | high |
| 75–100 | extreme |

where $PF_i$, $MPF_i$ and $W_i$ are probability factor, maximum probability factor and the weight of $i$th parameter, respectively. The calculated value enables investigators to assess the probability of ground instability occurrence (table 4).

Interpretation of ground instability probability in table 4:

low—low probability of ground instability, good conditions for mining operations;
medium—possible ground instability, medium conditions for mining operations;
high—high probability of ground instability, bad conditions for mining operations; and
extreme—extreme high probability of ground instability, extremely unfavourable conditions for mining operations.

## 4.2. Consequence of ground instability

Evaluation of the consequences causing both tangible and intangible losses or quantitative and qualitative losses is a very important step in the risk assessment [18]. The potential consequences of ground instability during CORP include:

(i) loss of coal production during production stoppage;
(ii) material cost to deal with the ground instability;
(iii) labour cost to deal with the ground instability;
(iv) the cost of repairing damaged equipment; and
(v) injuries to coal miners or fatalities.

Of the above five consequences, except for the injuries to the miners or fatalities, the remaining four can be explicitly measured by money. Therefore, all financial losses caused by ground instability during CORP can be calculated by adding the first four items together. To determine the economic loss, accident investigations were performed in a number of mines in Shendong Company. The average financial losses caused by the ground instability are as follows (at a profit of $30 per ton for thermal coal):

(i) loss of coal production during stoppage, per shift ~$200 000;
(ii) material cost to deal with the ground instability, per shift ~$2000;
(iii) labour cost to deal with the ground instability, per shift ~$1000; and
(iv) the cost of repairing damaged equipment ~$15 000.

Investigation cases include accidents lasting 1–20 days. The financial cost analysis of the ground instability demonstrates that the loss caused by coal production accounts for 92% of the total loss. In order to simplify the risk assessment process, the loss of ground instability can be determined based on the potential loss during coal production stoppage as given by the below equation:

$$L_{GI} = Q_{CP} \times N \times P_C, \tag{4.3}$$

where $L_{GI}$ is loss of coal production, $Q_{CP}$ stands for the quantity of planned coal production per shift, $N$ represents the number of shifts and $P_C$ indicates the profit of coal per ton in US dollars.

When assessing the losses caused by the ground instability, another important aspect is human casualties. Injuries to the personnel can be assessed using the method described in the Chinese standard GB/T 6441-1986 [43], which is commonly used in the Chinese mining industry. This method defines three types of injuries to personnel: minor injuries, severe injuries and deaths. Table 5 is the final evaluation of the consequence of ground instability, which combines personnel's and non-personnel's accidents.

**Table 5.** Matrix for assessing the consequences of ground instability.

| consequences for the miner owing to ground instability (according to GB/T 6441-1986) | financial loss owing to ground instability (equation (4.3)) | | |
|---|---|---|---|
| | <$200 000 | $200 000–600 000 | >$600 000 |
| minor injuries | low | medium | high |
| severe injuries | medium | high | extreme |
| deaths | extreme | extreme | extreme |

**Table 6.** Matrix of the assessment of ground instability risk during CORP.

| probability of the ground instability (table 4) | consequences of the ground instability (table 5) | | | |
|---|---|---|---|---|
| | low | medium | high | extreme |
| low | low risk | low risk | medium risk | high risk |
| medium | low risk | medium risk | medium risk | high risk |
| high | medium risk | medium risk | high risk | extreme risk |
| extreme | high risk | high risk | extreme risk | extreme risk |

Interpretation of ground instability consequences in table 5:

low—ground instability may cause no injuries or minor injuries; treatment of ground instability should not exceed one shift;

medium—ground instability may cause some injuries; average time for treatment of ground instability is approximately three shifts;

high—ground instability may cause severe injuries; average time for treatment of ground instability for this category might be more than three shifts; and

extreme—ground instability may cause fatalities.

## 4.3. Evaluation of ground instability risk

Sections 4.1 and 4.2 introduce the methods for calculating the probability and consequence of ground instability during CORP, respectively. The level of ground instability risk can be determined with a matrix as indicated in table 6.

Interpretation of risk levels of ground instability in table 6:

low risk—*acceptable risk*: no preventive measures are necessary. Constant monitoring of the level of risk is required to provide work-related safety in a given area;

medium risk—*acceptable risk*: mining operations can be carried out under strict supervision, and measures are taken considering the rules of optimizing costs;

high risk—*unacceptable risk*: control measures are required widely to reduce the level of risk and to improve safety of mining operations; and

extreme risk—*unacceptable risk*: mining operations cannot start before actions to reduce risk to the acceptable level are taken.

For a given longwall face and remnant pillar involved, loading parameters and transfer parameters are the true geological conditions and are known. Design parameters are the longwall designs according to those geological conditions and are unknown before the assessment.

When applying this method, an important issue is to select an appropriate initial value for unknown design parameters to start a risk assessment. To solve this problem, the initial values of the unknown parameters are set at the values corresponding to the lowest allowable cost. If the obtained risk level is not acceptable, the value of the unknown parameter needs to be changed to a lower risk level. This process can be repeated until the risk level is acceptable. For example, the initial values for parameters related to the hydraulic props can be set at the same values as in the other entries not affected by the overlying pillars. Figure 9 summarizes all the steps involved in the form of an algorithm.

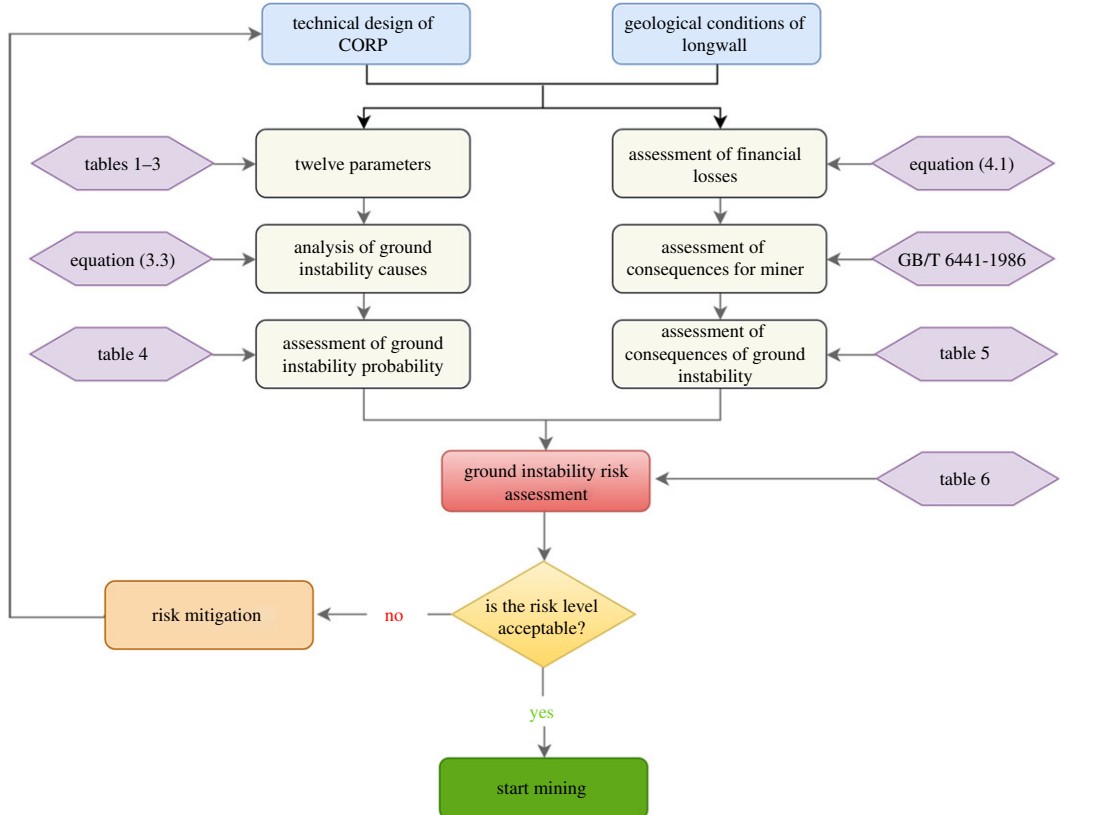

**Figure 9.** Algorithm for assessing the ground instability risk during CORP.

To apply a model, it must be ensured that it can accurately reflect the actual situation. Thirty cases of CORP from 20 coal mines were analysed using the proposed method (see table 11 in appendix B). The results confirm that the risk assessment results obtained from 27 working faces are consistent with the actual situation, with an accuracy rate of 90%. Among them, there are three failure cases. Owing to the existence of a fault within the coal pillar, which is not considered in the proposed method, the risk level of ground instability in Pingmei no. 1 mine is underestimated. The other two cases occurred in Xiegou mine and Tongxin mine, where longwall top coal caving (LTCC) method were used. Owing to the difference in the overburden movement between partial cutting mining (LTCC) and full cutting mining, the predicted risk levels of ground instability are inconsistent with the actual situation.

# 5. Application of the proposed method

## 5.1. Geological settings

In this section, the proposed risk assessment method is practically applied to lower-seam longwall 10–103 (LW10-103) of Mugua coal mine, China. The upper-seam (no. 9) has been mined out with a series of interpanel coal pillars left in the gob which affect the lower longwalls in seam no. 10. The interburden thickness varies from 2 to 20 m, which means that the upper-seam coal pillar may have an adverse effect on the lower working face. Therefore, it is necessary to assess the ground instability risk for longwalls in seam no. 10.

LW10-103 is approximately 250 m wide along the dip and 2000 m long along the strike as illustrated in figure 10. It is located beneath the upper-seam longwalls of 9-201, 9-203 and 9–205. During the advancement of LW10-103, a 20 m wide interpanel pillar will be encountered. Seam no. 10 averaged about 3 m in thickness, and its depth of cover was 290 m. The coal seam is nearly horizontal with a mean dip angle of 5°. The interburden between the two seams was 10 m and consisted of mudstone and sandstone. During the development of the roadway, the accumulated water in the upper gobs has been completely drained, and the working face is not affected by the overlying water. A summary of values of the parameters is presented in table 7. It should be noted that if the risk level of the mining

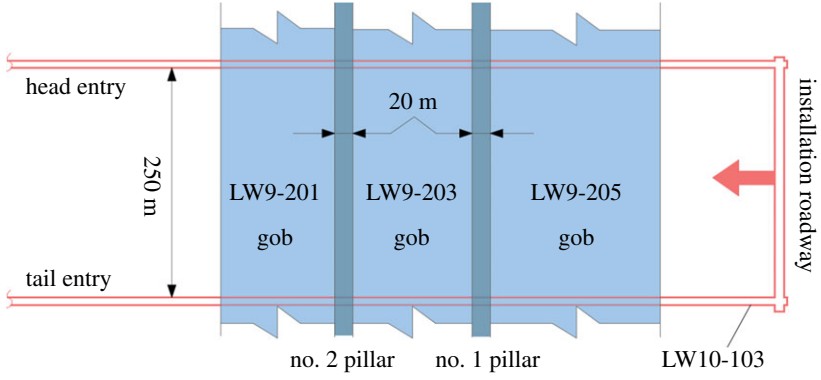

**Figure 10.** Layout of multiseam longwall panels in Mugua mine.

**Table 7.** Parameters and subgroups of LW10-103.

| no. | parameter | value | level |
|---|---|---|---|
| 1. | depth of cover | 290 | 1 |
| 2. | abutment angle | 20° | 2 |
| 3. | pillar type | isolated pillar (20 m) | 4 |
| 4. | interburden thickness | 10 m | 4 |
| 5. | interburden geology | 50% | 2 |
| 6. | overlying massive strata | not present | 0 |
| 7. | mining height | 3 | 2 |
| 8. | direction of mining | both involved | 4 |
| 9. | crossing angle | unknown | — |
| 10. | powered support in face | unknown | — |
| 11. | hydraulic props in entry | unknown | — |
| 12. | grouting of surrounding rock | unknown | — |

height is reduced to subcategory 1, coal resources will be greatly wasted. Therefore, the risk level of the mining height is medium.

## 5.2. Results

Eight of the 12 parameters in the proposed method are known, and the remaining four are unknown. The initial risk levels of the crossing angle, powered support intensity, support increasing ratio of the hydraulic props and surrounding rock are set at extreme (90°), high (0.6 MPa), extreme (1) and high (broken and not grouted), respectively. The risk assessment confirms that the risk level of ground instability is extreme, as shown in table 8.

Since entries perpendicular to the pillars have been developed, LW10-103 can cross the coal pillars by tilting the face, the risk level of which decreases to high. The risk level of the powered supports and hydraulic props can be gradually decreased from extreme to low. In addition, the risk level of the surrounding rock grouting was fixed at a high level, which can be adjusted later according to the actual state of the surrounding rock.

By repeatedly reducing the risk level of each unknown parameter, the resulting risk level has dropped from extreme to medium (table 8). Therefore, the mining plan for crossing coal pillars with a risk level of medium is determined as follows:

 (i) the crossing angle is inclined to 5°, which means that the head entry and tail entry are offset by 16 m during CORP;
 (ii) the support intensity of the powered supports is increased to 1.06 MPa corresponding to the working resistance of 7800 kN;

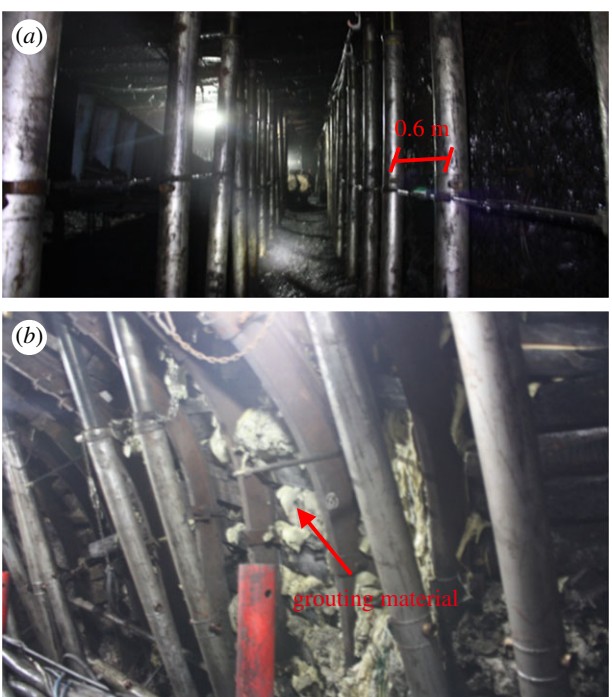

**Figure 11.** Photos of entry during CORP: (*a*) hydraulic props in the entry ahead of the face; (*b*) grouting of the broken surrounding rock.

**Table 8.** Design parameters optimization.

| no. | crossing angle $a/°$ | powered support ($n$/MPa) | hydraulic props ($L_p$ and $S_p$/m) | grouting | risk level |
|---|---|---|---|---|---|
| 1. | 90 | 0.97 | 20 and 1.2 | broken and not grouted | extreme |
| 2. | 85 | 0.97 | 20 and 1.2 | broken and not grouted | high |
| 3. | 85 | 1.16 | 20 and 1.2 | broken and not grouted | high |
| 4. | 85 | 1.49 | 20 and 1.2 | broken and not grouted | high |
| 5. | 85 | 1.72 | 20 and 1.2 | broken and not grouted | high |
| 6. | 85 | 1.72 | 40 and 1.2 | broken and not grouted | high |
| 7. | 85 | 1.72 | 40 and 0.6 | broken and not grouted | high |
| 8. | 85 | 1.72 | 50 and 0.6 | broken and not grouted | high |
| 9. | 85 | 1.72 | 50 and 0.6 | broken and grouted | medium |
| 10. | 85 | 1.72 | 50 and 0.6 | not broken | medium |

(iii)  the distance and row spacing of the hydraulic props are increased to 50 m and decreased to 0.6 m (two rows), respectively, as shown in figure 11*a*; and

(iv)  if the surrounding rock is indeed broken, grouting should be performed.

The medium risk does not mean that the surrounding rock is absolutely stable. In practice, the miners grout the coal wall ahead of the working face because it is broken, as shown in figure 11*b*. In the end area of the working face, the convergence of the roof and floor is relatively large, exceeding 0.3 m. By removing the debris on the floor and installing the bottom bolt, the operation of the equipment and the progress of the working surface was ensured. Eventually, LW10-103 successfully crossed two overlying coal pillars.

## 6. Discussion

Although there are six design parameters and six geological parameters, the weight of the parameters is different, that is, the weight of the geological parameters is 0.531, while that of the design parameters is

0.469, which reflects that geology is the dominant factor in the process of ground instability caused by CORP. Optimizing design parameters can only reduce the risk as much as possible, but cannot absolutely eliminate the danger.

In addition to the geology parameters, mining height is also sometimes unavailable for the risk control. In other words, mining height is rarely reduced in practice, although the reduction of mining height can alleviate the intensity of the surrounding rock movement. On the one hand, reducing the mining height leaves the top or bottom coal behind, which may not be conducive to roof management or equipment movement. On the other hand, some mine operators believe that when the powered supports are subject to high overburden loads, the height of the supports should be increased as much as possible [23]. The aim of this action is to provide sufficient space for the yielding and closure of the powered supports so that they will not be iron bound.

In recent years, some new technologies have been used to control the ground instability during CORP. One of the techniques is to perform presplitting blasting on the coal pillars, thereby eliminating their capacity to support the overburden and relieving the stress concentrations. Yuan *et al.* [6] analysed the blast hole arrangement, the calculation of the charge, etc., and conducted an industrial test in Nanliang coal mine. If the lower-seam working face is threatened by the massive overlying strata, the danger can also be eliminated by deep-hole presplitting blasting on the massive strata [44]. Moreover, some coal mines use special hydraulic supports instead of hydraulic props for the entry support ahead of the face, which greatly improves the stability of the surrounding rock. When these new technologies are employed during CORP, it is necessary to reasonably reduce the risk level of their corresponding parameters in the risk assessment. For example, if hydraulic supports are used in the entry support ahead of the face, their risk level can be adjusted to low.

# 7. Conclusion

Various types of difficulties resulting from overlying remnant pillars, including ground instability in the face and entries, pose a significant threat to mining operations, equipment and miners' health in the lower seam. Therefore, it is of great significance to develop a risk assessment method for ground instability during CORP so as to prevent such accidents.

In this work, a systematic risk assessment method is developed on the basis of extensive multiseam mining experience and the examination of the related literature. In this method, 12 effective parameters controlling ground instability, including three loading parameters, three transfer parameters and six design parameters, were identified. The weight of each parameter is obtained based on experts' experience, and the consequences of the ground instability were evaluated from two aspects, namely miners' health and financial losses. According to the probability and consequence of the ground instability occurrence, the ground instability risk can be assessed at four levels: low, medium, high and extreme.

This method enables the mining engineers who are not familiar with crossing coal pillars and cannot judge the risk based on their experience to use international experience for assessing the risk of ground instability. Furthermore, the main advantage of this method is that it can be used as a useful guideline for selecting reasonable powered supports, hydraulic props and crossing angles and for deciding whether to grout or not with a minimum risk of ground instability.

It should be mentioned that some new technologies, including the blasting of coal pillars and massive stratum and hydraulic supports for the entry support ahead of the face, have been employed to solve the ground instability during CORP. In future works, scholars should focus on the effects of applying these new technologies and consider whether to introduce them into risk assessment methods.

Data accessibility. This article has no additional data.

Authors' contributions. R.P. performed the data analyses and wrote the manuscript; Y.L. checked and revised the content about the abutment angle; H.W. and Y.X. collected some related cases and conducted in-depth analysis; H.Y. designed the questionnaire and calculated the parameter weights.

Competing interests. The authors declare that they have no conflict of interest.

Funding. This research was supported by the Graduate Research and Innovation Foundation of Chongqing, China (CYB18032), the National Natural Science Foundation of China (51904043, 51774059, 51764010, 51874109), Guizhou Province Science and Technology Support Program, China ([2019]2861) and Guizhou Province Excellent Youth Training Plan ([2019]5674).

# Appendix A

**Expert marking questionnaire for parameter weights of mining crossing overlying remnant pillars**

Name:_________________          Professional Title:_______________

Employer:______________          Position:____________________

Thank you for taking the time to fill out this questionnaire. The questionnaire aims to determine the parameter weights involved in mining crossing overlying remnant pillars. Please rate the importance of parameters based on your knowledge, experience and intuition in the field of multiseam mining.

**(1) Parameter system**

There are 12 parameters in total, including depth of cover, abutment angle, pillar type, interburden thickness, interburden geology, overlying massive strata, mining height, direction of mining, crossing angle, powered supports in the face, hydraulic props in entry and grouting of the surrounding rock. The detailed description of each parameter is omitted here (§3).

**(2) Marking criteria**

The 1–9 scale method is used to obtain mark results. The marking criteria are shown in table 9. By comparison, relative importance among different parameters can be determined by experienced field practitioners. Please fill in the marking results in table 10.

**Table 9.** Marking criteria.

| mark of $i$ compared to $j$ | equally important | weakly important | strongly important | very strongly important | extremely important |
|---|---|---|---|---|---|
| | 1 | 3 | 5 | 7 | 9 |

comment: 2, 4, 6, 8 are the intermediate values between two adjacency values; if mark $i$ to $j$ is $p_{ij}$, then mark $j$ to $i$ is $p_{ji} = 1/p_{ij}$

**Table 10.** Marking sheet.

| no. | parameter $i$ | parameter $j$ | mark of $i$ compared to $j$ |
|---|---|---|---|
| 1 | depth of cover | abutment angle | |
| 2 | depth of cover | pillar type | |
| 3 | depth of cover | interburden thickness | |
| 4 | depth of cover | interburden geology | |
| 5 | depth of cover | overlying massive strata | |
| 6 | depth of cover | mining height | |
| 7 | depth of cover | direction of mining | |
| 8 | depth of cover | crossing angle | |
| 9 | depth of cover | powered supports in the face | |
| 10 | depth of cover | hydraulic props in entry | |
| 11 | depth of cover | grouting of the surrounding rock | |
| 12 | abutment angle | pillar type | |
| 13 | abutment angle | interburden thickness | |
| 14 | abutment angle | interburden geology | |
| 15 | abutment angle | overlying massive strata | |
| 16 | abutment angle | mining height | |
| 17 | abutment angle | direction of mining | |

(*Continued.*)

| no. | parameter *i* | parameter *j* | mark of *i* compared to *j* |
| --- | --- | --- | --- |
| 18 | abutment angle | crossing angle | |
| 19 | abutment angle | powered supports in the face | |
| 20 | abutment angle | hydraulic props in entry | |
| 21 | abutment angle | grouting of the surrounding rock | |
| 22 | pillar type | interburden thickness | |
| 23 | pillar type | interburden geology | |
| 24 | pillar type | overlying massive strata | |
| 25 | pillar type | mining height | |
| 26 | pillar type | direction of mining | |
| 27 | pillar type | crossing angle | |
| 28 | pillar type | powered supports in the face | |
| 29 | pillar type | hydraulic props in entry | |
| 30 | pillar type | grouting of the surrounding rock | |
| 31 | interburden thickness | interburden geology | |
| 32 | interburden thickness | overlying massive strata | |
| 33 | interburden thickness | mining height | |
| 34 | interburden thickness | direction of mining | |
| 35 | interburden thickness | crossing angle | |
| 36 | interburden thickness | powered supports in the face | |
| 37 | interburden thickness | hydraulic props in entry | |
| 38 | interburden thickness | grouting of the surrounding rock | |
| 39 | interburden geology | overlying massive strata | |
| 40 | interburden geology | mining height | |
| 41 | interburden geology | direction of mining | |
| 42 | interburden geology | crossing angle | |
| 43 | interburden geology | powered supports in the face | |
| 44 | interburden geology | hydraulic props in entry | |
| 45 | interburden geology | grouting of the surrounding rock | |
| 46 | overlying massive strata | mining height | |
| 47 | overlying massive strata | direction of mining | |
| 48 | overlying massive strata | crossing angle | |
| 49 | overlying massive strata | powered supports in the face | |
| 50 | overlying massive strata | hydraulic props in entry | |
| 51 | overlying massive strata | grouting of the surrounding rock | |
| 52 | mining height | direction of mining | |
| 53 | mining height | crossing angle | |
| 54 | mining height | powered supports in the face | |
| 55 | mining height | hydraulic props in entry | |
| 56 | mining height | grouting of the surrounding rock | |
| 57 | direction of mining | crossing angle | |
| 58 | direction of mining | powered supports in the face | |
| 59 | direction of mining | hydraulic props in entry | |

**Table 10.** (*Continued.*)

| no. | parameter *i* | parameter *j* | mark of *i* compared to *j* |
|---|---|---|---|
| 60 | direction of mining | grouting of the surrounding rock | |
| 61 | crossing angle | powered supports in the face | |
| 62 | crossing angle | hydraulic props in entry | |
| 63 | crossing angle | grouting of the surrounding rock | |
| 64 | powered supports in the face | hydraulic props in entry | |
| 65 | powered supports in the face | grouting of the surrounding rock | |
| 66 | hydraulic props in entry | grouting of the surrounding rock | |

### (3) Other opinions

1. Your familiarity with the above content:

   A. unfamiliar
   B. slightly familiar
   C. familiar
   D. very familiar

2. The basis for judging the problem is

   A. theoretical analysis
   B. practical experience
   C. intuition

# Appendix B

See table 11 on the next page.

**Appendix B. Table 11.** Case summary.

| mine | face | IC1 /m | IC2 /° | IC3 | IC4/ m | IC5/ % | IC6 | IC7 /m | IC8 | IC9/° | IC10 | IC11 | IC12 | assessment results | actual production delay | reasonable assessment |
|---|---|---|---|---|---|---|---|---|---|---|---|---|---|---|---|---|
| Shigetai | 31201 | 138 | 13 | C | 41 | 32 | A | 4.1 | C | 0 | 1.9 | 2 | B | extreme | 60 days | yes |
| | 12102 | 66 | 13 | D | 5 | 35 | A | 2.8 | B | 0 | 2.0 | 2 | B | extreme | 2 days | yes |
| | 12103 | 64 | 13 | D | 3 | 35 | A | 2.8 | B | 0 | 2.0 | 2 | B | extreme | 2 days | yes |
| | 12105 | 79 | 13 | D | 10 | 36 | A | 2.8 | B | 0 | 2.0 | 2 | B | extreme | 13 days | yes |
| Daliuta | 22103 | 86 | 15 | D | 23 | 25 | A | 3.6 | C | 0 | 1.8 | 3 | B | extreme | 5 days | yes |
| | 52304 | 220 | 15 | D | 150 | 31 | A | 6.9 | B | 0 | 1.3 | 3 | A | low | 0 | yes |
| Bulianta | 22307 | 185 | 16 | D | 39 | 30 | A | 6.8 | A | 0 | 1.2 | 2 | B | moderate | 0.5 days | yes |
| Huojitu | 12304 | 98 | 10 | C | 19 | 31 | A | 4.5 | C | 0 | 1.3 | 1.5 | B | extreme | 2 days | yes |
| | 12305 | 116 | 10 | C | 19 | 26 | A | 4.6 | C | 0 | 1.5 | 1.5 | B | high | 1 days | yes |
| | 12306 | 97 | 10 | C | 21 | 49 | A | 4.7 | C | 0 | 1.5 | 1.5 | B | extreme | 2 days | yes |
| | 12313 | 103 | 10 | D | 13 | 35 | A | 4.7 | B | 0 | 1.5 | 1.5 | B | extreme | 2 days | yes |
| | 12314 | 86 | 10 | D | 3 | 34 | A | 4.7 | B | 0 | 1.5 | 1.5 | B | extreme | 3 days | yes |
| Shangwan | 12302 | 245 | 18 | D | 16 | 21 | A | 4.7 | B | 0 | 1.5 | 2 | C | moderate | 1 day | yes |
| Buertai | 42103 | 445 | 18 | C | 71 | 56 | A | 6.7 | C | 0 | 1.1 | 2 | C | low | 0 | yes |
| Yujialiang | 43309 | 150 | 18 | B | 21 | 36 | A | 2.1 | — | 90 | 2.7 | 3 | B | moderate | 0 | yes |
| Halagou | 12101 | 70 | 18 | B | 13 | 45 | A | 1.8 | C | 0 | 3.8 | 1.5 | B | moderate | 0 | yes |
| | 22207 | 80 | 18 | C | 25 | 40 | A | 5.0 | C | 0 | 1.4 | 1.5 | B | moderate | 0.5 days | yes |
| Duerping | 63405 | 240 | 21 | B | 6 | 40 | A | 3.3 | C | 0 | 1.2 | 2 | B | extreme | 4 days | yes |
| Tongxin | 8203 | 500 | 25 | B | 130 | 73 | D | 21 | C | 40 | 0.4 | 3 | B | extreme | 0.5 days | no |
| Yongdingzhuang | 8910 | 350 | 25 | B | 11 | 63 | A | 2.9 | C | 0 | 1.7 | 3 | B | high | 0.5 days | yes |
| Jinhuagong | 8707 | 350 | 25 | B | 20 | 65 | A | 2.6 | C | 0 | 2.0 | 3 | B | high | 0.5 days | yes |
| Pingmei no.1 | 31100 | 830 | 19 | B | 16 | 36 | A | 2.3 | C | 0 | 2.6 | 2 | C | moderate | 4 days | no |

(*Continued.*)

**Appendix B. Table 11.** (*Continued.*)

| mine | face | IC1 /m | IC2 /° | IC3 | IC4/ m | IC5/ % | IC6 | IC7 /m | IC8 | IC9/° | IC10 | IC11 | IC12 | assessment results | actual production delay | reasonable assessment |
|---|---|---|---|---|---|---|---|---|---|---|---|---|---|---|---|---|
| Sitai | 8206 | 220 | 25 | B | 15 | 58 | A | 3.1 | C | 45 | 1.7 | 1.5 | C | high | 6 h | yes |
| Silaogou | 8402 | 320 | 25 | B | 27 | 57 | A | 4.2 | C | 0 | 1.7 | 1.5 | C | moderate | 4 h | yes |
| Tongjialiang | 81015 | 210 | 25 | B | 17 | 75 | A | 3.6 | C | 0 | 1.5 | 4.5 | A | low | 0 | yes |
| Hanjiawan | 3301 | 127 | 14 | B | 23 | 36 | A | 2.8 | C | 0 | 2.8 | 2 | B | moderate | 0.5 days | yes |
| Xiegou | 23103 | 320 | 17 | B | 55 | 45 | A | 15 | — | 90 | 0.6 | 4.5 | B | high | 0 | no |
| Kaida | 1601 | 48 | 12 | D | 13 | 35 | A | 2.1 | B | 0 | 2.8 | 2 | B | extreme | 2 days | yes |
|  | 6⁻²305 | 43 | 12 | D | 15 | 26 | A | 1.6 | B | 0 | 3.6 | 1.5 | B | extreme | 30 days | yes |
| Pingsuo no. 2 | 1103 | 160 | 14 | B | 5 | 43 | A | 4.7 | — | 90 | 1.6 | 2 | A | low | 0 | yes |

Notes: IC3: A, yielded pillar; B, critical pillar; C, wide pillar; D, gob–solid boundary.
IC6: A, not present; B, present/less than 20 m; C, present/between 20 and 60 m; D, present/more than 60 m.
IC8: A, from gob to solid; B, from solid to gob; C, both involved.
IC12: A, not broken; B, broken and not grouted; C, broken and grouted.

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
