## [Reviewer comments · Royal Society Open Science]

Review History

RSOS-201249.R0 (Original submission)

Review form: Reviewer 1

Is the manuscript scientifically sound in its present form?

Yes

Are the interpretations and conclusions justified by the results?

Yes

Is the language acceptable?

Yes

Do you have any ethical concerns with this paper?

Yes

Have you any concerns about statistical analyses in this paper?

No

Recommendation?

Accept with minor revision (please list in comments)

Comments to the Author(s)

I am very impressed with this paper. I have read many high quality Chinese papers in recent years, but nearly all of them have been based on numerical modeling and ground monitoring data. This paper employs a different, empirical approach, which has been highly successful in US and Australian coal mines. The empirical method requires (1) a clear understanding of the factors that affect a specific strata control problem, and (2) Lots of case history data from operating mines. This paper does a good job of both.

My one major suggestion is that more information could be provided regarding the 30 case histories the paper mentions. How many were successful, and how many were not? What were the most important parameters that changed between them?

I also have several minor comments:

- 1) The caption to figure 2 has two mis-spellings: "Roof Fall" and "Disasters"
- 2) The narrative at the top of page 19 should be in past tense: "was ensured" and "crossed"
- 3) On page 20, "proves" should be changed to "reflects," and the awkward phrasing should be changed to "leaves the top or bottom coal behind."

Review form: Reviewer 2

Is the manuscript scientifically sound in its present form?

No

Are the interpretations and conclusions justified by the results?

No

Is the language acceptable?

Yes

Do you have any ethical concerns with this paper?

No

Have you any concerns about statistical analyses in this paper?

Yes

Recommendation?

Major revision is needed (please make suggestions in comments)

Comments to the Author(s)

1. Page 2 - Introduction, lines 57 to 60: Authors sentence "Field measurements and numerical modeling studies have shown that the vertical stress concentrations caused by the remnant coal pillars can propagate to the floor from a maximum depth of 120-200 m [8-10]."

In this sentence, I think author refers to interburden distance of 120-200m. This sentence needs to be rewritten.

2. Page 3 - Background, lines 49 to 52: Author said " For longwall extraction, the remnant pillars can generally be classified as either [24]:
Isolated pillars that are surrounded by the gob on two or more sides; or
Gob-solid boundaries with the gob on one side; "

Definition of the isolated pillars isn't correct. Mark et al. (<https://www.cdc.gov/niosh/mining/UserFiles/works/pdfs/aomss.pdf>) used the following

definition: A gob-solid boundary carries a single, distributed abutment load, while an isolated remnant pillar is subjected to two, overlapping abutments.

3. It is very hard to understand what is happening in Figure 1. Author should use a cross-section view similar to figure 7 in

<https://www.cdc.gov/niosh/mining/UserFiles/works/pdfs/aomss.pdf>

4. Page 4 – lines 8 to 10: Author says “The ground instability of longwall face and entry is usually the greatest hazard due to the load transfer from highly stressed remnant pillars.”

This sentence has to be rewritten or deleted. It is a vague sentence.

5. Page 4 – lines 32 to 41: Author says “Peng [27] concluded that the abutment angle ranges from 55° to 85° depending on the geological strength. Weak strata, such as shales and mudstones, cave in at a steeper angle than stronger strata such as sandstones and conglomerates. Therefore, abutment angle is divided into three categories: high, when it is between 55° and 65°; medium, when it is between 65° and 75°; and low, when it is between 75° and 85°.”

I think the author confuses abutment angle with “angle of influence” in subsidence (authors can refer to the Surface Deformation Prediction System Manual (Agioutantis, 2017) for the term). Abutment angles referred by author are extremely high and to my knowledge neither Dr. Peng or anyone else characterized abutment angle with strata caving yet. In the USA, average number of 21 is accepted as the standard abutment angle (Mark, 1992). Field measurements showed that value changes from 5 to 40 etc. (Colwell, 1999).

Author also treats abutment angle as a physical phenomenon which isn't. It is merely a conceptual model that represents the amount of mining induced loads on the abutments. Author can refer to publications of Dr. Christopher Mark.

Therefore, I believe technical information detailed in this section are wrong and author should completely revise this section.

6. Page 12 – lines 20 to 24 Author says “Due to lack of sufficient effective cases for quantitative risk assessment, this paper employs a semi-quantitative method to assess the risk of ground instability during crossing overlying remnant pillars. In fact, this method has been used by a large number of scholars to evaluate the roof fall risk in longwall mining and retreat mining [18,21].”

First of all, author only referred two scholars. This isn't a large number. It isn't clear how author come up with the probability factors and weights without a large database. Author indicated that 30 experts were surveyed to compare and score the relative importance of every two parameters and statistical analysis were used to compute final weights. Authors should explain the details of this survey since it is the most critical part of this research. All the other information is collected from the previous publications. What type of questions are asked to experts? Also, how are we gone judge the capacity of the expert if her or his expertise in the multiple seam mining? How are we gone judge if an expert is biased by her or his unique experience since most of the mining applications have unique geological and operational parameters?

I recommend major revision for this paper.

Decision letter (RSOS-201249.R0)

Dear Dr Pan

The Editors assigned to your paper RSOS-201249 "Assessment of ground instability risk of lower seam longwall panels during crossing overlying remnant pillars" have now received comments from reviewers and would like you to revise the paper in accordance with the reviewer comments and any comments from the Editors. Please note this decision does not guarantee eventual acceptance.

Please submit your revised manuscript and required files (see below) no later than 21 days from today's (ie 20-Aug-2020) date. Note: the ScholarOne system will 'lock' if submission of the revision is attempted 21 or more days after the deadline. If you do not think you will be able to meet this deadline please contact the editorial office immediately.

on behalf of Professor Zach Agioutantis (Associate Editor) and R. Kerry Rowe (Subject Editor)
openscience@royalsociety.org

Reviewer comments to Author:
Reviewer: 1

Comments to the Author(s)

I am very impressed with this paper. I have read many high quality Chinese papers in recent years, but nearly all of them have been based on numerical modeling and ground monitoring data. This paper employs a different, empirical approach, which has been highly successful in US and Australian coal mines. The empirical method requires (1) a clear understanding of the factors that affect a specific strata control problem, and (2) Lots of case history data from operating mines. This paper does a good job of both.

My one major suggestion is that more information could be provided regarding the 30 case histories the paper mentions. How many were successful, and how many were not? What were the most important parameters that changed between them?

I also have several minor comments:

- 1) The caption to figure 2 has two mis-spellings: "Roof Fall" and "Disasters"
- 2) The narrative at the top of page 19 should be in past tense: "was ensured" and "crossed"
- 3) On page 20, "proves" should be changed to "reflects," and the awkward phrasing should be changed to "leaves the top or bottom coal behind."

Reviewer: 2

Comments to the Author(s)

1. Page 2 – Introduction, lines 57 to 60: Authors sentence “Field measurements and numerical modeling studies have shown that the vertical stress concentrations caused by the remnant coal pillars can propagate to the floor from a maximum depth of 120–200 m [8-10].”

In this sentence, I think author refers to interburden distance of 120-200m. This sentence needs to be rewritten.

2. Page 3 – Background, lines 49 to 52: Author said “ For longwall extraction, the remnant pillars can generally be classified as either [24]:

Isolated pillars that are surrounded by the gob on two or more sides; or
Gob-solid boundaries with the gob on one side; “

Definition of the isolated pillars isn't correct. Mark et al.

(<https://www.cdc.gov/niosh/mining/UserFiles/works/pdfs/aomss.pdf>) used the following definition: A gob-solid boundary carries a single, distributed abutment load, while an isolated remnant pillar is subjected to two, overlapping abutments.

3. It is very hard to understand what is happening in Figure 1. Author should use a cross-section view similar to figure 7 in

<https://www.cdc.gov/niosh/mining/UserFiles/works/pdfs/aomss.pdf>

4. Page 4 – lines 8 to 10: Author says “The ground instability of longwall face and entry is usually the greatest hazard due to the load transfer from highly stressed remnant pillars.”

This sentence has to be rewritten or deleted. It is a vague sentence.

5. Page 4 – lines 32 to 41: Author says “Peng [27] concluded that the abutment angle ranges from 55 to 85° depending on the geological strength. Weak strata, such as shales and mudstones, cave in at a steeper angle than stronger strata such as sandstones and conglomerates. Therefore, abutment angle is divided into three categories: high, when it is between 55° and 65°; medium, when it is between 65° and 75°; and low, when it is between 75° and 85°.”

I think the author confuses abutment angle with “angle of influence” in subsidence (authors can refer to the Surface Deformation Prediction System Manual (Agioutantis, 2017) for the term). Abutment angles referred by author are extremely high and to my knowledge neither Dr. Peng or anyone else characterized abutment angle with strata caving yet. In the USA, average number of 21 is accepted as the standard abutment angle (Mark, 1992). Field measurements showed that value changes from 5 to 40 etc. (Colwell, 1999).

Author also treats abutment angle as a physical phenomenon which isn't. It is merely a conceptual model that represents the amount of mining induced loads on the abutments. Author can refer to publications of Dr. Christopher Mark.

Therefore, I believe technical information detailed in this section are wrong and author should completely revise this section.

6. Page 12 – lines 20 to 24 Author says “Due to lack of sufficient effective cases for quantitative risk assessment, this paper employs a semi-quantitative method to assess the risk of ground instability during crossing overlying remnant pillars. In fact, this method has been used by a large number of scholars to evaluate the roof fall risk in longwall mining and retreat mining [18,21].”

First of all, author only referred two scholars. This isn't a large number. It isn't clear how author come up with the probability factors and weights without a large database. Author indicated that 30 experts were surveyed to compare and score the relative importance of every two parameters and statistical analysis were used to compute final weights. Authors should explain the details of this survey since it is the most critical part of this research. All the other information is collected from the previous publications. What type of questions are asked to experts? Also, how are we gone judge the capacity of the expert if her or his expertise in the multiple seam mining? How are we gone judge if an expert is biased by her or his unique experience since most of the mining applications have unique geological and operational parameters?

I recommend major revision for this paper.

===PREPARING YOUR MANUSCRIPT===

- one version identifying all the changes that have been made (for instance, in coloured highlight, in bold text, or tracked changes);
- a 'clean' version of the new manuscript that incorporates the changes made, but does not highlight them.

This version will be used for typesetting if your manuscript is accepted.

===PREPARING YOUR REVISION IN SCHOLARONE===

Author's Response to Decision Letter for (RSOS-201249.R0)

See Appendix A.

Decision letter (RSOS-201249.R1)

Dear Dr Pan,

It is a pleasure to accept your manuscript entitled "Assessment of ground instability risk of lower seam longwall panels during crossing overlying remnant pillars" in its current form for publication in Royal Society Open Science. The comments from the Editors are included at the foot of this letter.

on behalf of Professor Zach Agioutantis (Associate Editor) and R. Kerry Rowe (Subject Editor)
openscience@royalsociety.org

Associate Editor Comments to Author (Professor Zach Agioutantis):

The authors have complied with the recommended changes. They have also used professional editing services to correct language and syntax.

Appendix A

Response

Dear Editors:

Thank you for your letter and for the reviewers' comments concerning our manuscript entitled "*Assessment of ground instability risk of lower seam longwall panels during crossing overlying remnant pillars*" (**RSOS-201249**). These comments are all valuable for revising and improving my paper.

The revision of the paper was assisted by other researchers. Yong Li helped me check and revise the content about the abutment angle (the 5th comment of reviewer 2). Hui Wang and Youlin Xu helped me collect some related cases and conduct in-depth analysis (the 1st comment of reviewer 1). Hongyun Yang helped me design the questionnaire and calculate the parameter weights (the 6th comment of reviewer 2). After careful consideration, I think they should be included as co-authors instead of in the acknowledgment section. This has also been approved by them. Therefore, I formally applied to list them as co-authors.

We have studied the comments carefully and have made corrections which we hope meet with your approval. Our responses to the reviewer's comments and modifications are detailed in the following pages. Line numbers (**highlighted** in the text) refer to the Manuscript with changes marked version.

In addition, a native speaker proofed the manuscript. Please refer to the certification listed as below.

Reviewer 1

1. My one major suggestion is that more information could be provided regarding the 30 case histories the paper mentions. How many were successful, and how many were not? What were the most important parameters that changed between them?

Response: Thanks for your advice. According to your advice, we have added the details of 30 cases, see Appendix B. The reasons for the three failed cases were also analyzed.

Answer:

Line 430-437

2. The caption to figure 2 has two mis-spellings: "Roof Fall" and "Disasters".

Response: Thanks for your advice. We have corrected these two mis-spellings in figure 2.

Answer:

Answer:

Line 108 (Fig. 2)

3. The narrative at the top of page 19 should be in past tense: "was ensured" and "crossed"

Response: Thanks for your advice. According to your advice, we have corrected the errors in English-language expression.

Answer:

Line 485

4. On page 20, "proves" should be changed to "reflects," and the awkward phrasing should be changed to "leaves the top or bottom coal behind."

Response: Thanks for your advice. According to your advice, we have replaced the term "proves" with "reflects" and changed the awkward phrasing to "leaves the top or bottom coal behind."

Answer:

Line 492, 498

Reviewer 2

1. Page 2 – Introduction, lines 57 to 60: Authors sentence “Field measurements and numerical modeling studies have shown that the vertical stress concentrations caused by the remnant coal pillars can propagate to the floor from a maximum depth of 120–200 m [8-10].”

In this sentence, I think author refers to interburden distance of 120-200m. This sentence needs to be rewritten.

Response: Thanks for your advice. According to your advice, we have rewritten this sentence.

Answer:

Line 45-48

2. Page 3 – Background, lines 49 to 52: Author said “For longwall extraction, the remnant pillars can generally be classified as either [24]:

Isolated pillars that are surrounded by the gob on two or more sides; or

Gob-solid boundaries with the gob on one side; “

Definition of the isolated pillars isn’t correct. Mark et al. (<https://www.cdc.gov/niosh/mining/UserFiles/works/pdfs/aomss.pdf>) used the following definition: A gob-solid boundary carries a single, distributed abutment load, while an isolated remnant pillar is subjected to two, overlapping abutments.

Response: Thanks for your advice. According to your advice, we have quoted the more reasonable definition you suggested.

Answer:

Line 85-86

3. It is very hard to understand what is happening in Figure 1. Author should use a cross-section view similar to figure 7 in

<https://www.cdc.gov/niosh/mining/UserFiles/works/pdfs/aomss.pdf>

Response: Thanks for your advice. We have redrawn the figure 1.

Answer:

Line 107 (Fig. 1)

4. Page 4 – lines 8 to 10: Author says “The ground instability of longwall face and entry

is usually the greatest hazard due to the load transfer from highly stressed remnant pillars.”

This sentence has to be rewritten or deleted. It is a vague sentence.

Response: Thanks for your advice. According to your advice, we have deleted this sentence.

Answer:

Line 97-98

5. Page 4 – lines 32 to 41: Author says “Peng [27] concluded that the abutment angle ranges from 55 to 85° depending on the geological strength. Weak strata, such as shales and mudstones, cave in at a steeper angle than stronger strata such as sandstones and conglomerates. Therefore, abutment angle is divided into three categories: high, when it is between 55° and 65°; medium, when it is between 65° and 75°; and low, when it is between 75° and 85°.”

I think the author confuses abutment angle with “angle of influence” in subsidence (authors can refer to the Surface Deformation Prediction System Manual (Agioutantis, 2017) for the term). Abutment angles referred by author are extremely high and to my knowledge neither Dr. Peng or anyone else characterized abutment angle with strata caving yet. In the USA, average number of 21 is accepted as the standard abutment angle (Mark, 1992). Field measurements showed that value changes from 5 to 40 etc. (Colwell, 1999).

Author also treats abutment angle as a physical phenomenon which isn’t. It is merely a conceptual model that represents the amount of mining induced loads on the abutments. Author can refer to publications of Dr. Christopher Mark.

Therefore, I believe technical information detailed in this section are wrong and author should completely revise this section.

Response: Thank you for pointing out an important mistake. Due to our carelessness, the definition of the abutment angle between figure 3 (from the vertical) and the text (from the mining horizon) is inconsistent. We have adopted a consistent definition (from the vertical) and modified other relevant places in the manuscript. It should be

pointed out that only the new abutment angle was changed to the complementary angle of the original abutment angle. This has no effect on the research content. We have completely revised this section.

Answer:

Line 141-151

6. Page 12 – lines 20 to 24 Author says “Due to lack of sufficient effective cases for quantitative risk assessment, this paper employs a semi-quantitative method to assess the risk of ground instability during crossing overlying remnant pillars. In fact, this method has been used by a large number of scholars to evaluate the roof fall risk in longwall mining and retreat mining [18,21].”

First of all, author only referred two scholars. This isn't a large number. It isn't clear how author come up with the probability factors and weights without a large database. Author indicated that 30 experts were surveyed to compare and score the relative importance of every two parameters and statistical analysis were used to compute final weights. Authors should explain the details of this survey since it is the most critical part of this research. All the other information is collected from the previous publications. What type of questions are asked to experts? Also, how are we gone judge the capacity of the expert if her or his expertise in the multiple seam mining? How are we gone judge if an expert is biased by her or his unique experience since most of the mining applications have unique geological and operational parameters?

Response: Thanks for your advice. Firstly, we have replaced the term “a large number of scholars” with “some scholars”. Secondly, we have added an explanation of how probability factors and weights are determined.

In addition, we have added a detailed questionnaire to the manuscript, see Appendix A, and added some information about the questionnaire survey and invited experts.

Answer:

Line 324, 330-333, 335-345